# Stag Beetle Elytra: Localized Shape Retention and Puncture/Wear Resistance

**DOI:** 10.3390/insects10120438

**Published:** 2019-12-05

**Authors:** Lakshminath Kundanati, Roberto Guarino, Nicola M. Pugno

**Affiliations:** 1Laboratory of Bio- Inspired & Graphene Nanomechanics, Department of Civil, Environmental and Mechanical Engineering, University of Trento, via Mesiano, 77, 38123 Trento, Italy; l.kundanati@unitn.it (L.K.); Roberto.guarino@psi.ch (R.G.); 2School of Engineering and Materials Science, Queen Mary University of London, Mile End Road, London E1 4NS, UK; 3Ket Lab, Edoardo Amaldi Foundation, Via del Politecnico snc, 00133 Rome, Italy

**Keywords:** beetle elytra, snap-through, puncture-resistant

## Abstract

Beetles are by far one of the most successful groups of insects, with large diversity in terms of number of species. A part of this success is attributed to their elytra, which provide various functions such as protection to their bodies from mechanical forces. In this study, stag beetle (*Lucanus cervus*) elytra were first examined for their overall flexural properties and were observed to have a localized shape-retaining snap-through mechanism, which may play a possible role in partly absorbing impact energy, e.g., during battles and falls from heights. The snap-through mechanism was validated using theoretical calculations and also finite element simulations. Elytra were also characterized to examine their puncture and wear resistance. Our results show that elytra have a puncture resistance that is much higher than that of mandible bites. The measured values of modulus and hardness of elytra exocuticle were 10.3 ± 0.8 GPa and 0.7 ± 0.1 GPa, respectively. Using the hardness-to-modulus ratio as an indicator of wear resistance, the estimated value was observed to be in the range of wear-resistant biological material such as blood worms (*Glyrcera dibranchiata*). Thus, our study demonstrates different mechanical properties of the stag beetle elytra, which can be explored to design shape-retaining bio-inspired composites with enhanced puncture and wear resistance.

## 1. Introduction

Insects represent the most successful group of organisms on Earth, accounting for almost 90% of all species, and are also highly diverse organisms. Their survival can be attributed to their small body, exoskeleton, adaptation to the changing environment, and (for some) the ability to fly. Of them, beetles represent 25% of all described species [1]. This extraordinary diversity is attributed to the beetles’ highly sclerotized strong fore wings which protect the hind wings when they are not in use and to their powerful flight using their membranous hind wings [2]. The forewings, termed as “elytra”, play an important role in the survival of beetles in battles and protection from the harmful environment outside. Elytra are multi-layered composites made of chitin fibrils and protein matrix [3], and have been examined for their microstructure and mechanical properties [4,5,6]. Elytra are biological composites of importance for bio-inspired designs because they inspire light-weight construction in buildings and light-weight camouflage materials [7,8]. It is crucial to perform a comprehensive mechanical characterization from a functional perspective.

In engineering structures, instabilities such as buckling and snap-through are usually considered as a disadvantage. We can find insects legs that are prone to failures by buckling as a dominant phenomenon [9], and some wasps have their egg laying needle-like organs that can prevent buckling failures [10]. In nature, thin sheets with curvature are common [11] and one example of a snap-through mechanism is the fast leaf closure of Venus flytrap plant [12]. Recently, novel ideas have been presented for the use of the above-mentioned mechanisms. For instance, the buckling phenomenon is exploited in the design of energy dissipation mechanisms such as absorbers, isolators, and stabilizers [13], and snap-through instability is used for amplifying the response of a fluid-based actuator [14]. A bi-stable mechanism is observed in beams that present two stable states, such as positions A and D (Figure 1A). In between, two intermediary states are observed in which snap-through and reverse snap-through mechanisms are activated during deformation, in reference to positions B and C, respectively. In displacement control conditions, the loaded beam traverses the path A–B–C–D (blue arrows). On the contrary, in force control conditions, the snap-through behavior is displayed by the path A–B–D (green arrows). This behavior allows the beam to absorb the released elastic kinetic energy described as “energy absorbed” (Figure 1B). In this phenomenon, shape change occurs either in unstable or stable manner and either reversibly or irreversibly. In such morphing structures, the shape change occurs either actively or passively depending on various factors such as intrinsic material behavior, geometry, temperature, load, and possibly some kind of actuation that enables switching between the two states [15]. Energy absorption designs that are aimed at preventing catastrophic failures use either microscopic or macroscopic deformation mechanisms. In biological materials such as bone, shape change is driven by the deformation at the micro-scale, such as delamination and micro-buckling of lamellar structures that are controlled by the distribution of tubules [16].

Protective armors are found across in many animals such as armadillos and turtles [17,18]. Armors are achieved by high wear and puncture resistance of the skin. The wear response of a biological material is dependent on the mechanical function it is used for, such as grasping, biting, or protecting the soft body [19]. For example, the components of the head articulation system of a beetle were observed to be in permanent contact and therefore should be resistant to wear [20]. We hypothesize that the beetles encounter some situations that could result in damage. For example, when the beetles fight and try to grip each other, there could be occurrence of point force and also scratching. Also, the beetles in their adult stage spend a significant amount of time underground before emerging out [21], subjecting the elytra to soil particle abrasion. So, it is of interest to examine their puncture and wear resistance. Elytra were shown to play an important function as a protective cuticle for the delicate wings and the abdomen of the Japanese rhinoceros (*Allomyrina dichotoma*) beetle [22]. There are a few works aimed at studying puncture-resistant mechanisms in fish scales and the cockroach abdomen cuticle [23,24].

Earlier studies measured the mechanical properties of elytra, but very few have tested the mechanical response of its whole structure. In this study, we demonstrated the snap-through and reverse snap-through mechanisms of the stag beetle elytra, which enable energy absorption during impacts and yet retain shape. Finite element simulations and analytical modelling of the snap-through phenomenon were performed to validate the experimental results. We also performed experiments to test protective capabilities using puncture tests and hardness measurements to comment on the wear resistance capabilities. To the best of our knowledge, this is the first time that such a snap-through mechanism has been highlighted in insect cuticle. We hope this study not only contributes to the existing knowledge but also to the development of bio-inspired shape-retaining composites with protective capabilities.

## 2. Materials and Methods

### 2.1. Sample Preparation

Male stag beetle samples were obtained in a dehydrated state from the collection of the MUSE Science Museum of Trento (Trento, Italy). Rectangular sections were cut out of the elytra separated from the insect (Figure 2A). The schematic of the insect abdomen cross-section region shows the stack up of the wing and elytra on the abdomen surface (Figure 2B). The sectioned samples were clamped to restrict the in-plane movement of the ends and actuation was done by using an acrylic blade to create a line contact on the external surface (Figure 2C). The acrylic blade was held by metal clamps that were connected to the load cell. In the whole elytra experiments, the samples were directly used after separating them from the insect body.

### 2.2. Optical and Electron Microscopy

The images of the insect and the microstructure of the elytra were captured using a Lynx LM-1322 optical microscope (Olympus corporation, Tokyo, Japan) and a charged coupled device (CCD) camera (Nikon, Tokyo, Japan) attached to the microscope. Scanning Electron Microscope (SEM) imaging of all the samples is performed after cleaning with the help of ultrasonication and a subsequent drying. The prepared elytra sections were carefully mounted on double-sided carbon tape and stuck on an aluminum stub, followed by sputter coating (Manual sputter coater, Agar Scientific Ltd., Stansted, United Kingdom) with gold. Images were captured using an SEM (EVO 40 XVP, ZEISS, Jena, Germany) with accelerating voltages between 5 and 20 kV. ImageJ software was used for all dimensional quantification reported in this study [26].

### 2.3. Mechanical Testing

All the mechanical tests were performed using a Messphysik MIDI 10 (MESSPHYSIK, Materials Testing GmbH, Genova, Germany) Universal Testing Machine and the forces were obtained using a ±10 N transducer (Leane International Srl, Parma, Italy). A qualitative approach was used to examine the overall elytra (24.8 × 10.2 mm) deformation by subjecting them to point loading. The objective was to try to simulate the loading from, e.g., the opponent male beetle mandible or impact loading during the fall from heights during battles on the trees (https://www.youtube.com/watch?v=0ND1JV_gs2M). Also, a goal was to demonstrate snap-through behavior qualitatively as seen in a living scenario, where the elytra are held in place with the help of the other elytron and the frictional locking elements, but are not fixed in a rigid way. So, we performed point-force compression tests on the whole elytra (two tests per single elytron) by placing it on a soft polymer substrate to mimic the softer abdominal structure and allow deformation without a constraint, at a rate of 0.01 mm/s. Bending experiments were performed on the mechanical cut samples (*n* = 2 samples, two tests each) to determine the snap-through behavior. These tests were performed on a custom-built bending set up machined out of hard plastic material (Figure 2C) with the possibility to clamp the sample edges. The rate of testing in three-point bending tests was kept at 0.01 mm/s. The puncture tests (*n* = 10, with two needles and five samples cut from each beetle) were performed using a 21G gauge needle that was connected to the same transducer (LEANE Corp., ±10 N) and the displacement (0.005 mm/s) was controlled using the same Messphysik MIDI 10 Universal Testing Machine. The samples (4 × 4 mm) were sectioned out to have minimal possible curvature so that it could rest on the PolyDiMethylSiloxane (PDMS) polymer substrate firmly and we did not observe any movement in the videos.

### 2.4. Finite Element Simulation

Elytra have a complex geometry and anisotropic material properties that are similar to those of composite laminates. We simplified the model for the finite element numerical simulations by considering the geometry as a simple curved plate, with average thickness *t* = 0.27 mm (also used the variation in thickness along the length), half-length *l* = 4.6 mm, half-width *w* = 1.98 mm, and maximum out-of-plane height *h* = 0.43 mm (with respect to the bottom surface), taken from the experimental setup. We assigned isotropic material properties, with Young’s modulus E = 650 MPa (i.e., around the average of the measured flexural moduli [25]). The values of Poisson’s ratio (ν = 0.1) and density (ρ = 1425 kg/m^3^) were chosen to be in the order of chitin properties [27] as a first approximation. For convenience, the simulations with orthotropic material properties were performed with a constant Poisson’s ratio along the three directions. We used a simplified symmetric model representing one-quarter of the complete geometry and specified two symmetry planes (Figure 3). Identical to the experimental setup, the model was clamped at both the ends and the load (*P*) was applied in displacement-control mode over a transversal line at the middle of the top surface. The total load was obtained following an incremental approach known as the Riks method [28]. The computational volume was discretized with 9000-s-order tetrahedral elements.

### 2.5. Nanoindentation

The dried elytra were sectioned in the thickness direction after embedding them in resin for two reasons. One is to fix the samples firmly and the other to polish for obtaining a flat surface. The use of resin is a standard practice in nanoindentation experiments and does not affect the results significantly. The embedded samples were polished using a series of 400-, 800-, 1200-, 2000-, and 4000-grade sand papers. Finally, the samples were polished using a diamond paste of particle sizes in the range of 6 µm and 1 µm, to obtain a surface of minimal roughness. The exocuticle region was carefully selected using optical microscope and the indentations were performed at this location. A Berkovich indenter was used to perform nanoindentation experiments employing a maximum load of 20 mN at the rate of 1200 mN/min. We performed a total of 12 indentations (with four indentations at each of the three different locations).

## 3. Results

### 3.1. Microstructure of the Elytra and the Abdominal Surface

We examined the structural features on the abdomen and the elytra that help in keeping them locked together, when the beetle is not in flight (Figure 4A). Our results show that elytra comprise primarily three different bulk layers. There is a top layer, which is exposed to the environment, followed by a middle layer, and a relatively thin layer that is connected to the middle layer by trabecular structures (Figure 4B). This is in agreement with the observations made in other beetle elytra [4]. The elytra and abdomen have primarily four contact regions that help in mitigating the forces. The first contact is the direct physical connection of the elytra to the thorax (C1, Figure 4A). The second contact is the locking mechanism between the left and right elytron that runs along the inside (C3, Figure 4A). These locking mechanisms were studied earlier and they require certain amount of force to decouple the elytra [29]. The third contact is a triangular-shaped structure (C2) which protrudes from the backside of the head, under which the top inner-side corners of the elytra get tucked in (Figure 4C,D). The fourth contact region (C4) is the abdomen edge surface (Figure 4C) regions with microtrachia that assist in friction-based locking (Figure 4E). Such microtrachia were also reported in the tenebrionid beetles [30]. Though some of these features were reported earlier in other beetle systems, we report them again here from *Lucanus cervanus* and also the other microstructures that possibly work together in keeping the elytra in contact with the abdomen firmly.

### 3.2. Deformation of the Elytra

Loading experiments were carried to examine the overall mechanical behavior of the whole elytra. The force-deflection curves from these experiments showed that the elytra deform elastically with a snap-through and reverse snap-through behavior. The shape changes were reflected in loading curves (Figure 5A), with a snap-through occurring at point B and reverse snap-through occurring just before reaching point C. The shape changes of the elytra after the snap-through and reverse snap-through were observed with a change in curvature (Figure 5B,C). It appears that the elytra work similar to the principle of bi-axial curvature, which is reported to be an efficient way of reinforcing thin sheet structures [11]. Attempts at fabricating morphing structures led to the development of a macro-fiber-based two-layer cross-ply composite that morphs from one shape to another shape with the help of piezoelectric actuation [31]. On the contrary, the elytra appear to have a structural design that can switch back to the undeformed state without any active actuation system. The snap-through behavior in multiple ply composites is dependent on various factors like differences in Young’s modulus along different directions, thickness of the ply, and coefficients of thermal expansion [32]. The small defects in the material can also influence morphing in one or other stable states [33]. The response of the elytra resembled the mechanical response of a monostable buckling structure which has the capability to reverse snap-through after the load removal [34]. We showed the elytra can retain shape after absorbing the impact energy using a snap-through and reverse snap-through mechanisms even in an unconstrained condition as in our experiments. We can thus claim that the elytra when constrained on the beetle body with the microstructural features (Figure 4) may also dissipate more energy through frictional unlocking.

To perform mechanical analysis of the snap-through behavior of the elytra using a standard procedure, we performed bending experiments by clamping the sectioned out rectangular samples at the ends. In clamped condition, the samples remained in the stable buckled state after removal of the load unlike the whole elytra samples. Force-displacement curves showed good repeatability in both the loading responses (Figure 6). The loading response in clamped condition showed that buckling occurs at a load of 0.37 ± 0.02 N and is higher than the load in the buckled state.

An analytical analysis was performed to describe the experimental results in Figure 6 based on the so-called Von Mises arch [35,36]. Elytra are characterized by a continuous geometry with structural imperfections and nonlinearities. Though the present shallow-arch model is an approximation, it has the capability to help in understanding the basic principles of snap-through instabilities. As shown in Figure 6 (inset), we approximate the elytron as two hinged bars loaded at the center, with α the initial angle with respect to the horizontal plane. Under the assumption of small angles, the load-displacement relation (Appendix A) is given by:(1)$$F\left(\delta \right)=k_{a}\delta \left(2\alpha^{2}-3\alpha \frac{\delta }{l}+\frac{\delta^{2}}{l^{2}}\right)+2k_{\theta }\frac{\delta }{l^{2}}$$
where *F* = load, *δ* = displacement, and *l* = half-length of the elytron, where $k_{a}$ is the axial stiffness of the equivalent bar and $k_{\theta }$ is the rotational stiffness of the lateral hinges. Here we can assume ka≈EAl=E 2wtl, where A is the cross section of the elytron, and we have used the half-length *l* instead of the actual curve length because of the small initial angle. In general, an increasing rotational stiffness produces an increase in the critical load as well as in the critical displacement; thus, all the curves tend to move towards the top right part of the diagram. The effect of the Young’s modulus, instead, is simply to increase the critical load without moving the position of the peak (i.e., the value of the critical displacement). Note that this reasoning is consistent with the buckling analysis of spherical shells, which can be considered an upper-bound estimate for our system and the details can be found elsewhere [37]. Also, in this case, the critical load is directly proportional to the Young’s modulus of the shell structure, but the shape of the snap-through curves is heavily dependent on the geometry. Considering a Young’s modulus of 1000 MPa, i.e., in the order of that measured value in a previous work [25], we can fit the average experimental curve with Equation (1), obtaining an axial stiffness, $k_{a}$ = 175 N/mm, and a rotational stiffness, $k_{\theta }$ = 5.2 Nmm/rad, for the equivalent shallow-arch model, with R2-value 0.856. As shown in Figure 6, the model is able to match quite well the initial part of the experimental load-displacement curve, while a larger deviation from experiments is observed in the post-critical stage. On the contrary, the post-buckling behavior can be captured sufficiently well by the finite-element simulations, with the simulated critical load in close agreement with the experimental data. Instead, the critical displacement value was found to be around half the corresponding experimental value.

The observed differences between experiments, theory, and simulations are due to the uncertainties in the geometry of the considered elytra and the presumably nonlinear material behavior, which do not allow us to make an extremely precise comparison. However, the simplified analytical model and the finite element simulations presented in this work still display a good potential in describing the observed experimental evidence. The variation is also attributed to assumption that the elytra is a homogeneous flat plate with unidirectional curvature as opposed to the bi-axial curvature and also their multilayered composite nature. This remains a limitation of the study but demonstrates the ability of simplified theoretical calculation to closely validate the experimental observation. Thus, elytra are structures with the ability to retain shape or recover from snap-through instantly, unlike the naturally existing Venus fly trap and the respective engineering composites. The multilayered structure of elytra appears to be similar to that of helmets that have a stiffer outer shell that enable impact distribution and thus avoiding concentrated loads [38]. Note that the experiments were carried out on naturally dried specimens because the selected species belongs to a near threated (International Union for Conservation of Nature (IUCN) Red list) state. Thus, some mechanical properties are expected to be higher than the cuticle in native state. The comparison of the experimental and analytical predictions remains still valid.

To understand the effect of dehydration of the cuticle, we performed additional simulations by changing the material properties such as the absolute value of the Young’s modulus and introducing orthotropic nature. Our results showed that there is relevant change only in the slope of the first part of the curve and the absolute value of the critical load, without significantly affecting the critical displacement. Also, the introduction of orthotropic material properties, which are closer to the actual material, does not change the shape of the force-displacement curve (Figure 7). Although there is a significant change in critical values of load and displacement, the phenomenon of snap-through remains, showing that the elytra can still undergo this mechanism to absorb energy even in their natural state.

### 3.3. Puncture and Wear Resistance

Puncture experiments were performed on the elytra using a standard surgical needle (Figure 8A), as explained in Section 2. Results from the puncture experiments showed moderate repeatability (Figure 8B). The puncture force was defined at the initial deflection in the force-displacement curve which results in cracking of the relatively harder surface layer (Figure 8B, denoted by a star sign). The average values of the puncture force were observed to be 1.8 ± 0.4 N (average over *n* = 10 tests). The variation in the force values can be attributed the regional variation in the microstructural features of the elytra, such as pore canals and trabecular structures. The penetration process is observed with the needle in contact with the top surface at the beginning and after the puncture (Figure 8C(i, ii)). The puncture force values were also in the order of recorded bite forces of stag beetles (6.9 ± 2.0 and 1.1 ± 0.4 N for males and females, respectively) [39]. The puncture force values of elytra are also similar to that of artificial gloves (HexArmor 7084, 9014, www.hexarmor.com) designed to resist puncture with average puncture force of 2.6 N and 3 N. The damage created from the puncture was shown in the image taken from the top surface of the sample, with brittle-like fracture of the top exocuticle layer (Figure 8D(i, ii)). Brittle behavior of the top layer (exocuticle) associated with the hardened nature of this layer is a consequence of cross-linking between helicoidally arranged nano-chitin fibrils and protein complex [40]. The primary resistance to puncture comes from the exocuticle, which also aids in improved wear resistance. In an earlier study, a correlation was observed with increased puncture resistance in the cuticle with higher sclerotization [41]. We can thus suggest that the force recorded in our studies will be close to that of the experiment performed on whole elytron. Similar puncture tests were performed on three species of cockroach cuticle, and puncture forces were measured to be between 1.2 to 30.7 N [24], though these cannot be directly compared due to the differences in needle characteristics.

In order to provide more accurate description of the puncture experiments, we have approximated the pressure during puncture using projected areas, by assuming the mandible as a spherical indenter and the surgical needle as a conical indenter (Figure 9A–C). The estimated pressure values show that elytra withstand higher pressures during puncturing than those experienced by the mandible bites (Table 1).

We measured the mechanical properties using indentation of the elytra external bulk layer to estimate the wear properties using the embedded cross-section samples, and the obtained force-depth curves appeared repeatable (Figure 10A,B). The Young’s modulus and hardness of this layer were found to be 10.3 ± 0.8 GPa and 0.7 ± 0.1 GPa, respectively. These values were in the same range of hardest exocuticle layer (E~10 GPa, and H~0.4 GPa) of the Sun beetle (*Pachnoda marginata*) cuticle in the head articulation system [19] and close to the values (H~0.95 GPa) of crab claw in the exocuticle region [42]. Earlier studies report that the dehydrated cuticle hardness is an order of magnitude higher as compared to that of the natural state, specifically in endocuticle and meso-cuticle [43]. The effects were significantly lower in exocuticle, with a factor of 2.4 in the values of E and 1.6 in the hardness values. However, the measured hardness and Young’s modulus values can be considered for the following estimation of wear and this variation is minimized because of the considered hardness to modulus ratio. The wear resistance can be roughly related with the ratio (H^1.5^/E) [44], which is calculated to be 0.05 ± 0.01 GPa^1/2^. These values were close to the values observed in Glycera jaw (~0.077 GPa^1/2^) which is known to be wear-resistant [45] and the high abrasion-resistant (0.06–0.08 GPa^1/2^) outer layer of the spider fang [44]. Materials with a hard-external layer and relatively softer inner layers enable them to be puncture-resistant and also flexible [46]. Thus, we can see from the above results that the elytra are designed to be both puncture- and wear-resistant. Though our studies are performed on dehydrated samples due to the nature of species selection, we can say that the estimated properties such as wear-resistance and puncture force are of the same order as compared to the hydrated state.

Currently, there is no scientific evidence of elytra undergoing snap through mechanism when a stag beetle or any other beetle falls on its back. Thus, we can only comment on the likely hood of the snap-through mechanism in living conditions. To interpret a situation of stag beetle elytra being subjected to impact loading at the end of a fall from a tree, we performed a simple theoretical analysis. We estimated the terminal velocity of the beetle in a free fall, equating gravity force (*F_g_*) and drag force (*F_d_*). It is used to calculate the maximal kinetic energy (*E_k_max_*) that the beetle builds up just before the impact. The kinetic energy values are then compared with the energy absorbed (*E_a_*) by the whole elytra during deflection, using the force (*F*)-deflection (*x*) curves. Also, we have performed another calculation to assess the situation of beetles falling with an optimal velocity (*v*) that is lower than terminal velocity and the corresponding impact energy becomes equal to the amount of energy that can be absorbed by both the elytra, using Equation (7).
(2)$$F_{d}=\frac{1}{2}C_{d}\rho v^{2}A$$
(3)$$F_{g}=mg$$
(4)$$v_{max}=\sqrt{\frac{2mg}{C_{d}\rho A}}$$
(5)$$E_{k\_max}=\frac{1}{2}mv_{max}^{2}=mgH_{max}$$
(6)$$E_{a}=\int Fdx$$
(7)$$2E_{a}=\frac{1}{2}mv^{2}=mgH$$
where *v_max_* = terminal velocity, *ρ* = density of the air (1.25 kg/m^3^), *A* = approximated surface area of the beetle (3 cm^2^), *m* = mass of the dehydrated beetle (1.7 g), *H_max_* = maximum height to reach terminal velocity, *H* = height to reach optimal velocity, and *g* = acceleration of gravity and $C_{d}$ = drag coefficient (1.04, from [47]). The estimated terminal velocity and maximum kinetic energy (*E_k_max_*) of the falling beetle during impact are 9.3 m/s and 72.6 mJ, respectively (Table 2). Also, we estimate the height of fall to reach terminal velocity and the height of optimal velocity to be, 4.4 m and 0.35 m, respectively.

Earlier studies recorded the falling behavior of insects such as ants and spiders. Their results show that the insects use a gliding mechanism to change the free-falling trajectory and achieve controlled landing back on the trees without hitting the ground [48,49,50]. The velocities of these beetles are more than twice the values reported in falling ants and locusts [48,49,50,51]. Also, the average body mass of these beetles is relatively much higher when compared to that of the ants and the spiders used in the experiments. Thus, it is less likely that stag beetles achieve gliding, especially in uncontrolled falls, and the overall impact force will be high enough to possibly cause damage to the cuticle. The estimated energy ($E_{a})$ that can be absorbed by a single elytron was approximately 2.9 ± 0.3 mJ, when the beetle falls on the backside and elytra together can absorb a significant portion of the total impact energy. 

## 4. Conclusions

In this study, we explored the mechanical properties of the stag beetle elytra which were not addressed in earlier studies and which would aid in performing different functions. Our experimental results showed that elytra are structures that can deform using a unique snap-through mechanism that enables energy absorption during impacts and can also retain their shape after impacts. Our finite element simulations showed that the elytra can undergo deformation using a snap-through mechanism, even if it has lower mechanical properties as compared to the properties reported in our dehydrated specimens. We also showed that the elytra support additional functions, such as puncture resistance and wear resistance. Due to the limitation of not being able to perform field studies to document snap-through mechanism, we performed a theoretical analysis to comment on the possibility of such phenomenon occurring. Such studies will help in designing of future bio-inspired materials and structures which could not only be shape-retaining but also wear- and puncture-resistant. Future studies, aimed at understanding the elytra layer–layer interface mechanisms under large deformation and using fresh samples, would aid in a better understanding and in the design of special application composites.

## Figures and Tables

**Figure 1 insects-10-00438-f001:**
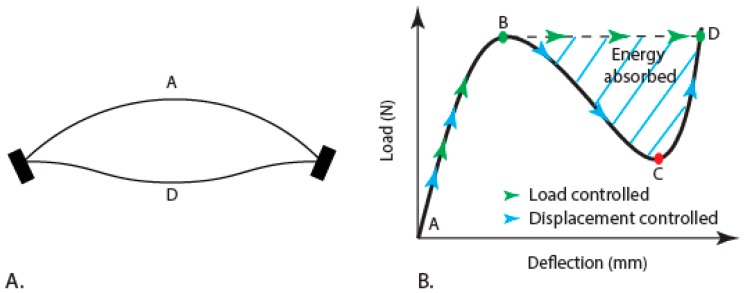
(**A**) Schematic of a shallow structure in the initial and final configuration. (**B**) The corresponding response is shown in the load-deflection curve (adapted from [13]).

**Figure 2 insects-10-00438-f002:**
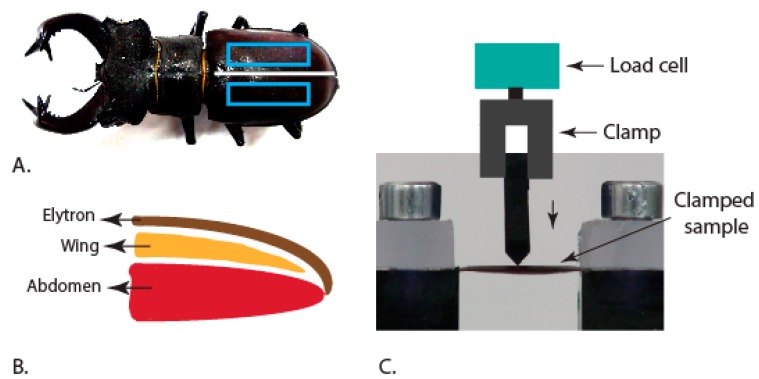
(**A**) Sample extraction location for the experiments (blue boxes) [25]. (**B**) Schematic of cross-sectional view of the insect body along the axis of symmetry of the body (white line in A). (**C**) Clamp set-up used for performing the experiments.

**Figure 3 insects-10-00438-f003:**
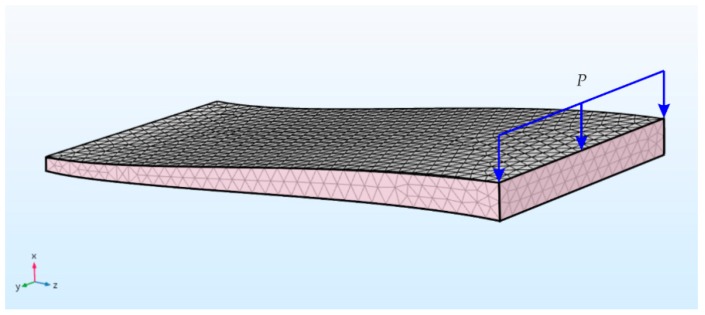
Meshed model and coordinate reference system, with highlighted the two symmetry planes (pink) and the applied edge load (*P*).

**Figure 4 insects-10-00438-f004:**
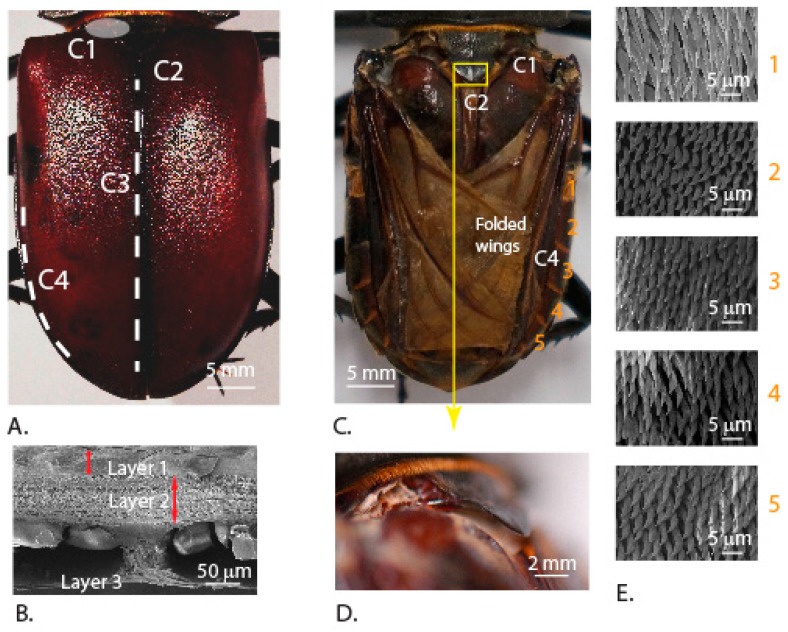
(**A**) Primary contact regions of elytra with abdomen. (**B**) Scanning electron micrograph of the cross-section of an elytron highlighting the different layers [25]. (**C**) Abdomen surface with folded wings after removal of elytra showing the hinge location (C1), a triangular shaped structure (C2) which accommodates the inner-side corners of elytra, and five surfaces with microtrachia (C4). (**D**) Magnified image of C2. (**E**) Scanning electron micrographs of the locking edge surfaces showing the microtrachia.

**Figure 5 insects-10-00438-f005:**
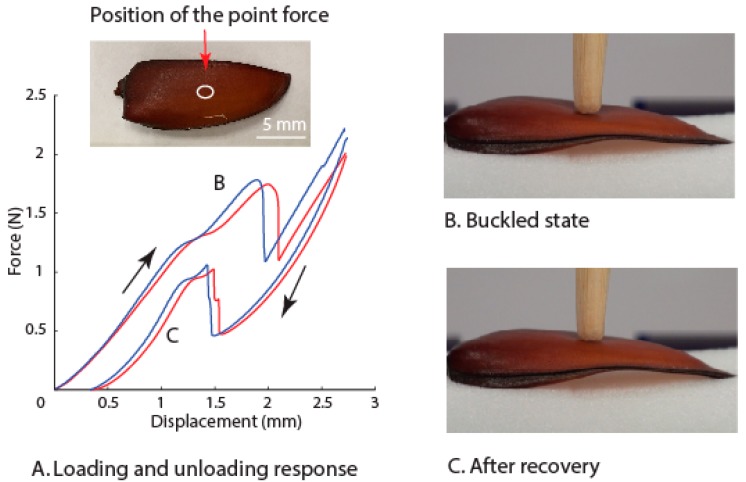
Whole elytra deformation response during loading and unloading. (**A**) Force-displacement curves and optical image showing the position of applied point force. (**B**) Elytron shape in the buckled state. (**C**) Elytron shape recovery after reverse snap-through.

**Figure 6 insects-10-00438-f006:**
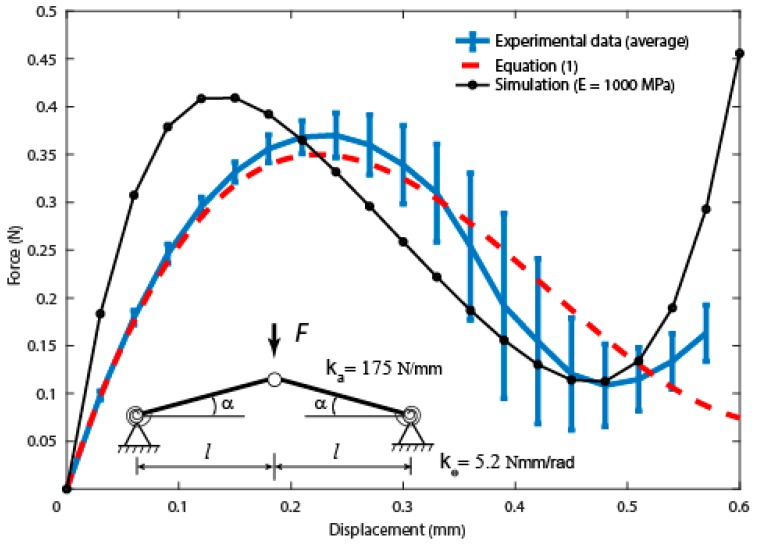
Comparison of sectioned elytra force-displacement curves from the experiments, analytical estimation, and the finite element simulation.

**Figure 7 insects-10-00438-f007:**
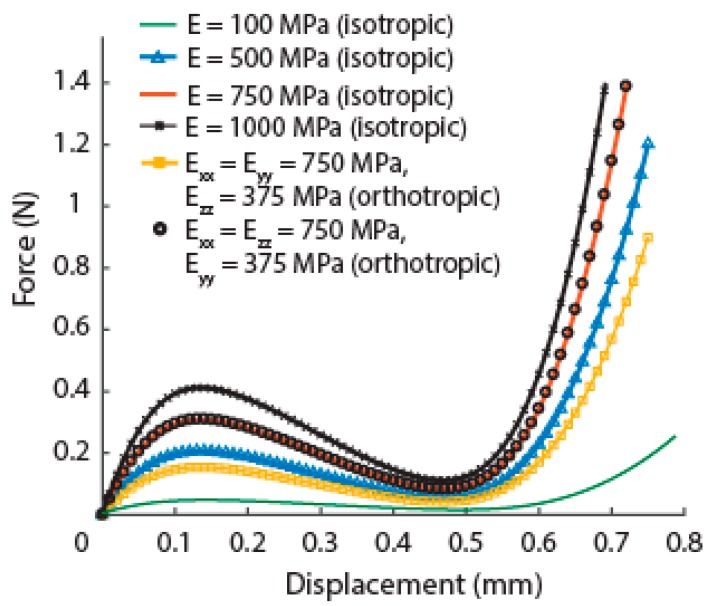
Force-displacement curves from finite element simulations for different material properties.

**Figure 8 insects-10-00438-f008:**
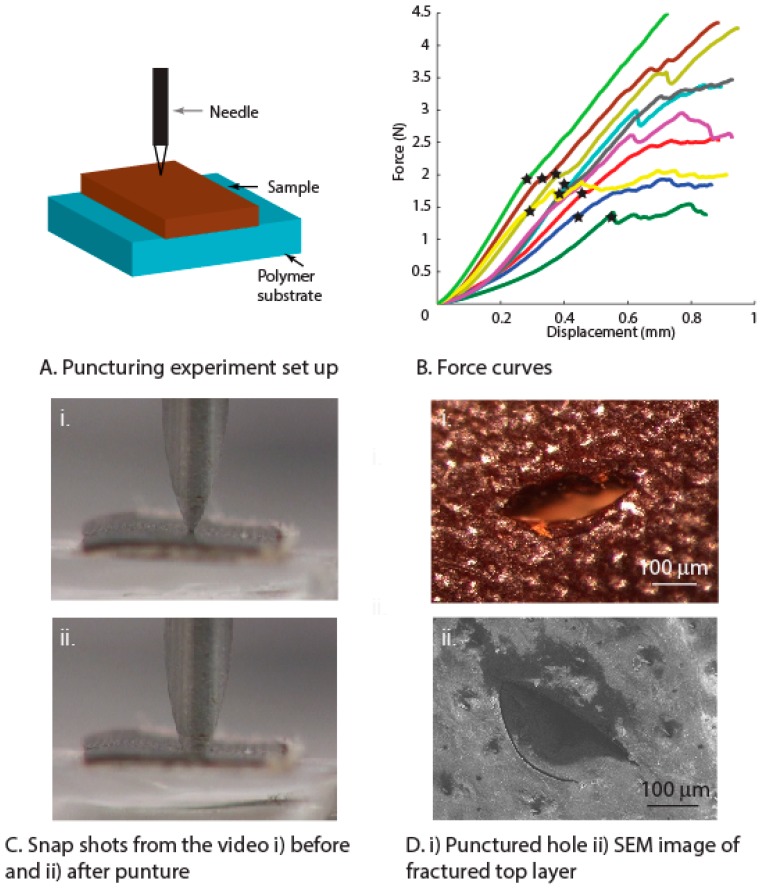
(**A**) Schematic showing the experimental set up used for puncture tests. (**B**) Force curves from the puncture tests, showing the puncture force (highlighted by *). (**C**) Snapshots from the video during puncturing. (**D**) (**i**). Beetle top surface with the punctured hole showing the brittle nature of the top layer; (**ii**). Brittle nature of the top layer observed from an SEM image of the punctured surface.

**Figure 9 insects-10-00438-f009:**
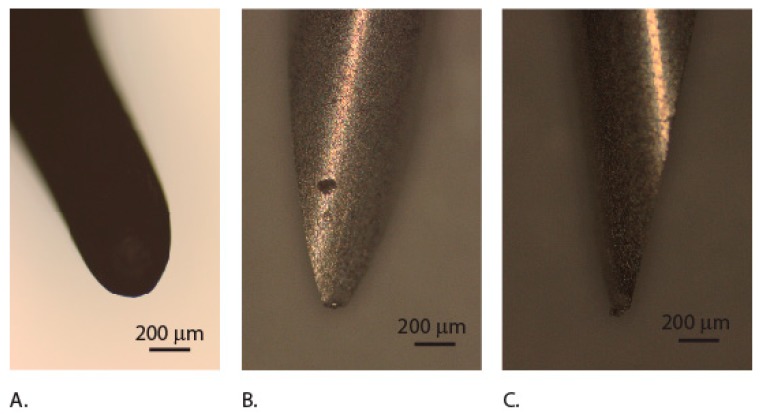
Images of the mandible. (**A**) Dark field image of mandible tip, and needle. (**B**) Front view. (**C**) Profile view.

**Figure 10 insects-10-00438-f010:**
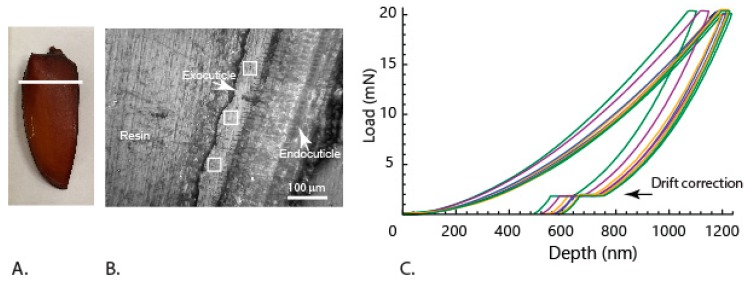
Optical image of (**A**) section location (white line). (**B**) the polished surface with the cuticle cross-section showing the selected locations of the nanoindentation on the external exocuticle, and the (**C**) force depth curves obtained from the experiments.

**Table 1 insects-10-00438-t001:** Puncture force presented as a pressure.

	Tip Diameter (µm) Measured at 100 µm from the Tip	Average Puncture Force (N)	Puncture Pressure (kPa)
**Mandible**	338	6.9	64
**Needle**	180	1.8	597

**Table 2 insects-10-00438-t002:** Estimated average values of velocities, heights of fall, kinetic energy of a falling beetle and the energy absorbeded by elytra.

Terminal Velocity (*v_max_*) (m/s)	Height to Reach Terminal Velocity, *H_max_* (m)	Optimal Velocity (*v*)	Height to Reach Optimal Velocity, *H* (m)	Maximum Kinetic Energy (*E_k_max_*) (mJ)	Energy Absorbed by Elytra, (2*E_a_*) (mJ)
9.3	4.4	1.8	0.35	72.6	5.8

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
