# Peer review of "Stag Beetle Elytra: Localized Shape Retention and Puncture/Wear Resistance"

_insects, 2019, doi:10.3390/insects10120438_

Round 1

Reviewer 1 Report

The authors have satisfactorily addressed most of my previous comments. As I read through the manuscript again, I found some more areas of concern, all minor, that I've highlighted in the document. The biggests of these is reporting puncture resistance as a pressure, not a force.

The biggest concern of mine at this point is that the authors do not explain the likelihood of the snap through mechanism in nature. Does this happen to beetles in the wild? If not ever documented, the manuscript loses purpose and the wear/puncture resistance becomes the headline story. 

Reviewer 2 Report

The manuscript is significantly improved after revision, thus, I recommend it for the publication. 

Author Response

We would like to thank the reviewer once again for the useful suggestions.

Reviewer 3 Report

The authors have addressed the comments and questions accurately, and the updated manuscript provides a more clear report of the work.

Author Response

We would like to thank reviewer once again, for the useful comments and suggestions.

This manuscript is a resubmission of an earlier submission. The following is a list of the peer review reports and author responses from that submission.

Round 1

Reviewer 1 Report

Comments on the manuscript by Kundanati et al

The authors are reporting the mechanical response of the elytra for stag beetles which are known for their battle behavior using their puncturing mandibles. In this work, they are reporting a snap-through mechanism which helps to dissipate a significant amount of energy induced from external contact loads. I found the study quite interesting and the experiments are well-designed to address the questions regarding the deformation mechanisms. Here, I have some comments and questions regarding the study:

- The fixation of the samples in buckling and puncture tests are not described. Have the samples been glued on top of a polymeric substrate? If there was no fixation, how authors ensured that the force-displacement curves (Figures 5 & 8) were not affected by the free movement of the samples (squeezing toward the substrate from the sharp corners of the cut edges)?

- In puncture experiment, since the sample size are kind of small, have authors noticed any correlation between sample size and force-displacement behaviors? In this regard, puncture experiments while the elytron is not dissected from beetle can provide further insights (or maybe it provides more realistic puncture forces).

- In page 9, line 296, “indentation properties” should change to “mechanical properties using indentation” since indentation cannot be a property of materials.

Reviewer 2 Report

This paper presented the special mechanical properties of the stag beetle elytra to mitigate the external forces using snap-through mechanisms. The finding in this paper is helpful for the reader.  
However, there are some concerns that need to be addressed.

1.     Why did the authors consider the dehydrated wing instead of the alive wing? In nature, the readers want to know how the elytra absorb energy from external forces.

2.     The information of the experimental setup and sample preparation in details are needed such as the size of the wing, the position of the point force, etc...

3.     There are no clear mechanical testing conditions. The authors could add the illustrations.

4.     Two samples for bending experiments is too small, please add more data to draw the conclusion more confidently.

5.     The authors could compare the obtained results of puncture forces with the bite forces of the stag beetle. Why did the author compare the measured puncture with that of the artificial gloves?

6.     Provide the position in the wing for the nanoindentation test because the mechanical properties of the wing are varied over the wing.

7.     Please provide the force-displacement curves in the nanoindentation test. For the reviewer opinion, it is not proper to compare the Young’s Modulus and hardness in this study with those in the reference [17] because the beetle in [17] is alive. Moreover, the condition for dehydration in this study is not clear for example how many days for the dehydration….

8.     For the simulation, the authors considered the orthotropic properties of the wing. How did the authors determine the Poisson’s ratio in x, y, z directions?

9.     In Figure 8D, please provide the photo with the clear punctured hole as well as the fractures near the hole.

Reviewer 3 Report

The manuscript entitled “Stag beetle elytra: localized shape-retaining, puncture and wear-resistant” investigates the mechanical properties of dehydrated Lucanus cervus elytra. The article is generally easy to follow and reasonably complete, but not particularly well-written as pointed out below.  There are numerous problems with the English, so of which I point out and others that should be fixable at the editing stage. The greatest weaknesses lie in the construction of the Introduction and presentation of the results. Below I present Major and Minor issues in the manuscript.

Major Issues:

The Introduction meanders around buckling topics without setting up the relevance of the manuscript’s results, particularly in a biological setting. This is a bio journal, not an engineering journal, after all. The topics includes are only tangentially pertinent to the work at hand. For example, do we care about wear resistance if studying the buckling mechanism and puncture resistance of elytra? At this point the reader does not know wear resistance will be covered in the results section.

Did the authors create Figure 1B? If so, there should be some physical basis for the shape, i.e. analysis or data. If not, it should be cited.

Sample preparation: dehydrated samples were used. Can the authors be certain that dehydrated samples have the same mechanical behaviors as in vivo elytra? You may not be able to get live beetles Lucanus cervus beetles, but you certainly can test a more common beetle in both hydrated and dehydrated states to compare. Using FE to justify use of dehydrated samples is not ideal when more common beetles could be used.

Why were N=2 measurements taken on each elytra? N=3 is the standard.

Nanoindentation: Will the resin affect the indentation values? What was the purpose fo the resin?

Why do the plots in Figures 5 and 6 look so different?

Is it possible to related the force curves in Figures 5 and 6 to energy dissipation as championed in the Introduction and Figure 1B?

The sentence that begins on line 203 is problematic. Higher deformation resistance does not directly translate the to more energy absorption on the part of the beetle. Yes, stiffer materials take more energy to deform, but if undeformed, there is not energy absorption. Stainless steel, for example, would not offer the beetle more absorption than their elytra because a fall would not generate enough force to significantly deform the steel.

I would expect the simulation and data to be more well-aligned. There are a number of ways the authors can more accurately represent the elytron in FE simulations by using laser scanning to generate a 3D model of the samples. What other attempts were made to make the FE more physically relevant? Only in the conclusion is it made clear what value FE has beyond comparing differing values of modulus, which is tenuous because the FE and data are so different. 

As the authors admit, puncture resistance of dehydrated elytra are likely quite different than those of in vivo, or recently extracted samples. As such, what is the significance of penetration resistance measured for dehydrated samples?

Conclusions: “We also showed that the elytra support additional functions, such as puncture resistance and wear resistance.” Without a baseline for comparison this conclusion is meaningless. Everything has some wear resistance, even the most delicate skin.

Minor Issues:

This article does not start strong. The first line of the abstract contains and error. Line 13: Beetles are not a species. 

Line 15: What harmful environmental factors?

Line 18: Insects are so lightweight, they are impervious to falls from any height, regardless of elytra strength. This is true for ANY insect. This notion should be removed from the manuscript.

Lines 21-22: The wording of this sentence is poor.

Line 23 needs a ‘respectively.’

Line 24-25: What other wear-resistant biomaterials? This could mean anything!

Line 33: Not all insects can fly.

Line 33-34: 'Of them, beetles are the most 33 diverse group of species of insects.’ This is a bold statement. Diverse in what sense? The statement is also completely devoid of citation. 

Lines 39-41: This sentence is 100% opinion. It should be reworded to state why these bio-composite may be of importance to bio-inspired designs.

Lines 49-50: It is not clear how displacement control and load control conditions are defined, or their relevance to buckling wings. The discussion of composites following does not seem to be very relevant to the topic of the paper.

Line 76: How do the authors define very few?

Line 124: e.g.
